# Psychological Interventions for Children with Autism during the COVID-19 Pandemic through a Remote Behavioral Skills Training Program

**DOI:** 10.3390/jcm11051194

**Published:** 2022-02-23

**Authors:** Flavia Marino, Paola Chilà, Chiara Failla, Roberta Minutoli, Noemi Vetrano, Claudia Luraschi, Cristina Carrozza, Elisa Leonardi, Mario Busà, Sara Genovese, Rosa Musotto, Alfio Puglisi, Antonino Andrea Arnao, Giuliana Cardella, Francesca Isabella Famà, Gaspare Cusimano, David Vagni, Pio Martines, Giovanna Mendolia, Gennaro Tartarisco, Antonio Cerasa, Liliana Ruta, Giovanni Pioggia

**Affiliations:** 1Institute for Biomedical Research and Innovation (IRIB), National Research Council of Italy (CNR), 98164 Messina, Italy; flavia.marino@cnr.it (F.M.); paola.chila@irib.cnr.it (P.C.); chiara.failla@irib.cnr.it (C.F.); roberta.minutoli@irib.cnr.it (R.M.); noemi.vetrano@irib.cnr.it (N.V.); claudia.luraschi@irib.cnr.it (C.L.); cristina.carrozza@irib.cnr.it (C.C.); elisa.leonardi@irib.cnr.it (E.L.); mario.busa@irib.cnr.it (M.B.); sara.genovese@irib.cnr.it (S.G.); rosy.musotto@irib.cnr.it (R.M.); alfio.puglisi@irib.cnr.it (A.P.); antoninoandrea.arnao@cnr.it (A.A.A.); giuliana.cardella@irib.cnr.it (G.C.); francescaisabella.fama@irib.cnr.it (F.I.F.); gaspare.cusimano@irib.cnr.it (G.C.); david.vagni@cnr.it (D.V.); gennaro.tartarisco@cnr.it (G.T.); giovanni.pioggia@cnr.it (G.P.); 2Azienda Sanitaria Provinciale, U.O.C. Neuropsichiatria Infantile, 91100 Trapani, Italy; pio.martines@asptrapani.it (P.M.); giovanna.mendolia@asptrapani.it (G.M.); 3S’Anna Institute, 88900 Crotone, Italy; 4Pharmacotechnology Documentation and Transfer Unit, Preclinical and Translational Pharmacology, Department of Pharmacy, Health Science and Nutrition, University of Calabria, 87036 Arcavacata, Italy

**Keywords:** autism, telehealth, behavioral skills training, parent training

## Abstract

COVID-19 has impacted negatively on the mental health of children with autism spectrum disorder (ASD), as well as on their parents. Remote health services are a sustainable approach to behavior management interventions and to giving caregivers emotional support in several clinical domains. During the COVID-19 pandemic, we investigated the feasibility of a web-based behavioral skills training (BST) program for 16 parents and their children with ASD at home. The BST parent training package was tailored to each different specific behavioral disorder that characterizes children with ASD. After training, we found a significant reduction in the frequency of all the targeted behavioral disorders, as well as an improvement in psychological distress and the perception of the severity of ASD-related symptoms in parents. Our data confirm the efficacy of remote health care systems in the management of behavioral disorders of children with ASD, as well as of their parents during the COVID-19 pandemic.

## 1. Introduction

The COVID-19 pandemic has had a profound impact on the health of children with autistic spectrum disorders (ASD) and their parents. In fact, these children have been identified as part of a group at higher risk of medical complications and social distress [1] from COVID-19. Moreover, parents of children with ASD often have trouble accessing behavioral services for their children [2]. This common condition has been significantly exacerbated by the COVID-19-related containment measures, with the risk of increasing lifelong impairments and comorbidities related to this disorder. More accessible interventions are urgently needed to support the families of children with ASD by promoting new techniques that can also facilitate clinical and supportive interventions at a distance.

Some studies have discussed the possible consequences the COVID-19 pandemic could have on individuals and their parents. Research suggests that people with autism are particularly vulnerable to conditions of prolonged isolation, as they have to adapt to new routines which can negatively affect their progress [3,4], and an online survey found that people with ASD exhibited an increase in problem behavior during lockdown periods [5].

To manage the spread of COVID-19, since March 2020, an international effort has been made to adapt all healthcare services to work “remotely by default” [6]. Telehealth has therefore become of primary interest, continually evolving to encompass new approaches, new clinical demands, and digital developments.

In several clinical domains, remote digital health services foster patient–clinician relationships, ensure continuity in treatments, and support families by simultaneously reducing burdens on health systems [7]. Technology-mediated care includes live video calls, monitoring health status by medical devices, e-mail, audio, and instant messaging. This service connects clinicians virtually with patients or caregivers by removing any physical distance. Remote health services have also been used to support families of children with ASD. Hyman et al. [8] clarified the definitions of various health services by establishing the validity of this method in a similar way to traditional face-to-face clinical settings, although evidence-based protocols are lacking. 

In this neuropsychiatric domain, remote health services may be more effective than in-person meetings. In fact, children with ASD have shown a positive sensitivity to the novelty of this method and the physical separation may allow clinicians to perform more naturalistic observations of the family setting [9]. Technologies related to remote health services also offer a cost-effective solution for extending the reach of behavioral interventions to families who do not live near a qualified provider, thus addressing inequalities in access to health care [10,11,12]. This method also has the advantage of easily training parents to be effective behavior-analytic teachers of their children [13]. Through technology-mediated care systems, it is thus possible to promote new kinds of online parent coaching by providing “anytime, anywhere” assistance to a parent who has access to the Internet during the pandemic era [14].

In this study, our aim was to demonstrate the feasibility and efficacy of a new web-based training approach aimed at reducing the frequency of targeted behavioral disorders in ASD children and improving parents’ reported sense of competence related to child-related behavioral dysfunctions during the lockdown. Our online parent program was focused on behavioral skills training (BST). This is a teaching procedure that involves the use of instructions, feedback, modeling, and rehearsal [15]. This method has been used to develop new ways to support children with ASD with specific problem behaviors. The intervention typically includes modeling and prompting procedures [16] in the context of function-based treatments [17], and functional analyses [18]. The effectiveness of the BST procedure has been widely demonstrated in several clinical contexts [15,16,17,18], although its implementation in remote health service programs has been poorly investigated. 

## 2. Materials and Methods

### 2.1. Enrollment

Twenty-eight parents of young children with autism were enrolled in the study. Parents were recruited and tested at the clinical facilities of the Institute for Biomedical Research and Innovation of the National Research Council of Italy (IRIB-CNR) in Messina and at the Centre for Autism Spectrum Disorders, Child Neuropsychiatry Unit, Provincial Health Agency of Trapani, Italy.

Inclusion criteria for the parents were: (1) being a native Italian speaker; (2) being biological parents; (3) having a home internet connection; and (4) being able to use web-based and telehealth tools. The inclusion criteria for ASD children were as follows: (1) being over 3 years of age; (2) clinical diagnosis of ASD based on DSM-5 criteria by a licensed neuropsychiatrist with the support of the Autism Diagnostic Observation Schedule, second edition (ADOS-2, Module 3); (3) a verbal development and performance quotient greater than 45; (4) no hearing, visual or physical disability preventing participation; and (5) not being on psychopharmacological treatment. All participants had had a previous diagnosis which was further confirmed through the evaluation and consent of experienced professionals from the research group (i.e., a child neuropsychiatrist and a clinical psychologist). Data were collected from October 2020 to May 2021 during the second and third waves of the pandemic in Italy. 

### 2.2. Ethics

All subjects provided informed consent for inclusion prior to their participation in the study. The study was conducted according to the guidelines of the Declaration of Helsinki and approved by the Committee of the Research Ethics and Bioethics Committee (http://www.cnr.it/ethics, accessed on 17 December 2021) of the National Research Council of Italy (CNR) (Prot. No. CNR-AMMCEN 54444/2018 01/08/2018) and by the Ethics Committee Palermo 1 (http://www.policlinico.pa.it/, accessed on 17 December 2021) of Azienda Ospedaliera Universitaria Policlinico Paolo Giaccone Palermo (report No. 10/2020–25/11/2020).

### 2.3. Study Design

The objective of the study was to assess the effectiveness of tele-assisted BST for parents in reducing the frequency of specific problem behaviors in children with ASD. To this end, given the wide behavioral variability and the absence of a control group a between-group design was not feasible and a multiple single subject design was also not feasible due to the wide range of techniques used and difficulty in assessing protocol adherence, fidelity and gathering data on a daily basis. Therefore, we applied a repeated measures design with four-time steps, using the frequency of undesired behavior as the outcome variable.

### 2.4. Treatment

The protocol was conducted through a web platform [G-Suite; Google LLC; Mountain view, CA, USA)] that gave access to video-conferencing tools. The Teleconsultation Center at IRIB-CNR in Messina had one teleconferencing workstation, with a basic webcam and headset, while parents at home were equipped with a tablet which was used to receive parent training and to record sessions for subsequent data coding and analysis (see Section 2.5 and Section 2.7). 

Parents and therapists briefly met via videoconferencing or telephone before and after each training session as needed in order to review the procedures, prepare the room and materials, discuss the results obtained, and to plan for the subsequent week’s session.

While both parents were expected to participate in the training, video-recording of the target behaviour and behavioural data gathering was only mandatory for one parent.

### 2.5. Protocol Phases and Parent Training Procedures

The experimental protocol consisted of a total of eight phases (see Table 1) divided into 13 sessions/meetings lasting 45 min each, in accordance with the ABA procedure, with the participation of both parents and children. The first four sessions were of pure training, eight central sessions were for active treatment, and the last session was for feedback. The final target was to match the treatment to the identified functions of problem behavior.

During Phase 0, an online meeting was arranged with families, and the therapists informed the parents of the study’s aims and procedures. Parent-report questionnaires were given to parents and gathered at the beginning of phase 1.

In Phase 1, the first operative meeting was scheduled. The therapist collected information on the individual child’s behaviors and discussed them with the parents. At the end of the meeting, the therapist provided them with an individualized frequency checklist in order to have an operationalized definition of the problem behaviors reported by the parents. Identified behaviour must have been observable, measurable, and repeatable. In the following week, the parents observed and monitored their child’s problem behaviors whilst completing the behavior frequency checklist.

In Phase 2, the second operative meeting was scheduled. The therapist and the parents discussed the results reported in the checklist and selected the target behaviour. During the online meeting, the therapist established a first baseline related to the problem behavior. The behavior selected for treatment was the more frequently observed behavior that reduced learning opportunities or social inclusion or was either physically or emotionally harmful for the child or their family. At the end of the meeting, the parents were instructed to record videos of the target behavior during the following week.

In Phase 3, the therapist, during the third meeting, compiled the Antecedent-Behavior-Consequence (ABC) worksheet using information collected through the recorded videos. Parents received live coaching on functional analysis procedures in order to understand and identify the antecedents [events that precede] and consequences [events that follow] of the problem behavior using the recorded videos as examples. The therapist provided instructions on filling out the Antecedent-Behavior-Consequence (ABC) worksheet. At the end of the meeting, the use of short videos helped parents to interpret the problem behavior and to complete the ABC worksheet with the assistance of the therapist. The therapist then asked the parents to record the data on the ABC worksheet during the following week while continuing to record the videos. 

In Phase 4 (fourth meeting), the therapist focused on analyzing the ABC worksheet compiled by the parents in order to identify the function of the monitored behavior through the recorded videos. Through a careful analysis of the antecedents and the consequences of the problem behavior over time, the therapist explained to parents why and how to follow the correct behavioral procedures that would favor a positive change. At the end of the meeting, parents were instructed to continue monitoring the behavior, recording video data, and filling out the ABC worksheet. Parents were now instructed to record videos with the same duration (45 min), in the same home location, and in the same context (i.e., presence/absence of parents) in order to gather the baseline (T0) for the BST.

In Phase 5, the fifth online meeting was held. The BST approach was integrated into the meeting sessions. This training utilizes instructions, modeling, rehearsal, and feedback in order to teach a new skill. The therapist first explains the skill to the parent, then models his/her behavior, who in turn models that of their child. The rehearsal phase is associated with role-playing used to train parents and model their behaviour during online meetings. At the end of each instance of the BST, the operator provided feedback on the performance. Feedback is positive when parents perform in accordance with the BST procedure, or corrective when the parents have difficulty in following the instructions correctly. In our study, we used both types of feedback to aid parents in following the BST instructions. Online coaching was complemented by homework with written instructions on the procedure, which was sent to parents at the end of meetings.

In Phase 6, a debriefing session was held between the therapist and the parents regarding the tasks that had been previously assigned. The procedural fidelity was evaluated in order to assess the adherence to the treatment and to maintain it if the behavioral change was moving towards the expected goal. In the case of lapses in the procedures, phase 5 was briefly resumed and the fidelity reestablished.

During the 6th to 12th online meetings, parents and therapists moved between phases 5 and 6 according to the training needs and continued to record videos.

Parents were instructed to continue to apply the procedures explained during BST and record a new session with the same duration (45 min), in the same home location, and in the same context (i.e., presence/absence of parents) every day. The idea was to reproduce the same procedures presented by the therapist during the sessions, leading to comparable outcome measurements over time. These sessions at home without a consultant’s input were recorded digitally using video-conferencing software for data collection. Specific reminders (by email or phone calls) were delivered to the parents in order to ensure that the videos were recorded during T1, T2 and T3.

Phase 7 was dedicated to monitoring fidelity and analyzing the progress of tasks. The results were analysed through a visual inspection of recorded materials during T0, T1, T2 and T3. Two additional experienced behavioral consultants, blind to any other result, evaluated the frequency of the problem behavior and the fidelity of the parental behavioral procedure. During the video analysis, the behavioral consultant marked the presence/absence (using “+” or “−” signs on a datasheet) of the problem behavior, obtaining a trend of the behavior over time, and marked the correct/incorrect procedure performed by the parents according to the schedule given by the therapist. Fidelity was dichotomized using “p” or “n” letters for each instance (pass or no-pass).

In the final phase (Phase 8), the feedback from the participants and parent-reported questionnaires was gathered. A timetable of the protocol, treatment and phases is reported in Table 2.

### 2.6. Training Experience

The therapists who delivered the interventions were all chartered psychologists, or psychotherapists, with behavioral analyst training and at least 5 years’ experience in working with children on the autism spectrum.

### 2.7. Outcome Measurements

Outcome measurements were divided into: (a) objective measurement of problem behavior in children occurring on the day immediately after each session; and (b) psychological assessment of parents recorded before and after treatment. 

For children with ASD, the main outcome was the frequency of the problem behavior which was recorded for statistical purposes at four different timepoints: Baseline (T0), and 20 days (T1), 40 days (T2), and 60 days (T3) after training (phase 5). 

In terms of the parents, we collected outcome measurements on parenting stress together with a scale for assessing the parent’s perception of the child’s behavioral manifestations. This evaluation was carried out twice, at the baseline and at the end of treatment. The main outcome measures were the Parenting Stress Index (PSI) to assess the level of stress before and after treatment and the Home Situation Questionnaire (HSQ-ASD), which provides objective measures of the perception and influence of children’s behavior on the parents’ lives.

#### 2.7.1. Parenting Stress Index/Short Form (PSI-SF)

The PSI-SF is a self-assessment questionnaire [19]. It takes about 10–15 min to complete the questionnaire. Parenting stress levels are assessed by analyzing three different factors: 1. characteristics of the children, 2. characteristics of the parent, and 3. aspects related to the parental situation. The short module is made up of 36 items, divided into three subscales: (1) Parenting Distress (PD), referring to the feelings of the parents; (2) dysfunctional parent–child interaction (P—CDI), which focuses on the child’s perception as unresponsive to parental expectations; and (3) Difficult Child (DC), which focuses on some of the characteristics of the child that make him/her easy or difficult to manage.

#### 2.7.2. Home Situation Questionnaire (HSQ-ASD)

The HSQ-ASD [20] is a caregiver-rated scale designed to assess the severity of disruptive and non-compliant behaviors in children. The scores obtained with this scale refer to the parent’s perception of their child’s behavioral manifestations. Within the scale, data are collected on the inflexibility (HSQ-I) and avoidance (HSQ-A) manifested by the child. This modified and revised version for ASD consists of 27 elements. Parents were asked to indicate whether their children had problems with compliance in these situations and, if so, to rate the severity on a Likert scale of 0 to 9, with higher scores indicating greater non-compliance.

### 2.8. Statistical Analysis

Statistical analyses were performed using SPSS v. 23.0 (IBM, Armonk, NY, USA). The Kolmogorov–Smirnov test was carried out which confirmed the assumptions of normality only for psychological outcome measures.

Families who dropped out and families who continued with treatment were compared using a *t*-test for continuous variables and chi-square test for categorical variables.

For children, although the sample size was small, based on our experience in treating behavioral disorders in young children with ASD, we expected large effects. Therefore, we applied repeated measures ANOVA, reporting the post hoc power observed. The Greenhouse–Geisser correction was used if conditions of sphericity were not met.

The initial frequency of behavior can differ considerably for different children and different behaviors. For each participant, we therefore used the ratio of frequencies measured at different time steps with the initial frequency.

Inter-rater agreement was computed using Cohen’s kappa (*k*).

For the parents’ measures (PSI-SF; HSQ-ASD), we expected a smaller effect size, and therefore used non-parametric (Wilcoxon signed-rank test) and parametric statistics (paired *t*-test) aimed at analyzing the effects of the treatment on parents’ outcome measures. We adjusted the alpha level using a Šidák correction for hierarchical multiple comparisons, with alpha = 0.025 for the primary measures (SI/SF and HSQ-ASD scores).

## 3. Results

The attrition rate was 42%. Indeed, six families dropped out before the beginning of phase 1, due to difficulties in managing weekly online connections, work commitments and/or family management of other children. Eight children with ASD and their relative parents (n°16), completed the treatment and were finally analyzed. Table 3 shows the demographic and psychological characteristics of children with ASD and their parents. The between-groups comparison showed no significant differences on any descriptive and psychological characteristic (*p* > 0.05).

Table 4 shows each individual behavioral symptom which was intended to be eliminated or reduced and its function. In Table 5, the behavioral treatment for each case is presented through the web-based training protocol with a brief explanation. The different procedures were performed as reported by Cooper and colleagues [18]. As expected, children with ASD showed heterogeneous behavioral spectrum disorders. Treatments were tailored to each specific behavioral symptom with the aim of reducing its frequency. 

### 3.1. Fidelity

No parent reported with videos every day. On average, 2.93 (0.73) [2.10–4.32] videos were reported per week. There were no weeks with less than one video for each participant. No video was missing at T0, T1, T2 and T3. For all the families, behavioural data were gathered only by the main caregiver (in all families, it was the mother).

Procedural fidelity of the parent was 0.81 (0.13) [0.75–1.00] at T1, 0.97 (0.07) [0.80–1.00] at T1, 1.00 (0.00) [1.00–1.00] at T3. It should be noted that the number of instances on which fidelity was based decreased with the increase in time-steps.

### 3.2. Inter-Rater Agreement

Inter-rater agreement was excellent. Raters agreed in 94% of 224 instances for behavioral frequency, *k* = 0.883, and 92% of 119 instances for parents’ procedural fidelity, *k* = 0.849. No disagreement was present in instances at T3. For the other time-steps, a final agreement was reached in all cases through discussion between the raters.

### 3.3. Behavioural Results

During active treatment, there was an immediate trend towards a better clinical outcome (Table 6), which at the end of the treatment was significantly improved (Figure 1).

Mauchly’s test of sphericity was not significant, Mauchly’s W = 0.440, *X*^2^ = 4.70, *df* = 5, *p* = 0.460.

The multivariate test was significant, *F* (3, 5) = 247, *p* < 0.001, n^2^ = 0.993 with an observed power (OP) of 1.00. The within-subject effect was also significant *F* (3, 21) = 92.3, *p* < 0.001, n^2^ = 0.930, OP = 1.00. However, the between-subject effects were also significant, *F* (1, 7) = 398, *p* < 0.001, n^2^ = 0.983, OP = 1.00.

In order to better understand the structure of change, we tried different polynomial models of frequency-change in time, and the best fit was for a linear model with *F* (1, 7) = 770, *p* < 0.001, n^2^ = 0.991, OP = 1.00. 

A decrease in behavioral frequency was present in all children in all the time steps, and a pairwise comparison showed that differences were significant between T1 and T0 (M = 38% [17%; 58%], *p* = 0.002), T2 and T1 (M = 26% [5%; 47%], *p* = 0.016), and T3 and T2 (M = 26% [2%; 50%], *p* = 0.032).

### 3.4. Parental Wellbeing

We also evaluated the effects of our web-based training on the psychological wellbeing and perception of severity in parents of ASD children, measured before and after the treatment (Table 7). As concerns the PSI-SF scale, we found a significant reduction in total psychological stress and PD and P-CDI subscales after treatment. A similar beneficial effect was also detected in the HSQ-ASD scale, where parents showed a significant reduction in the perception of the severity of inflexibility (HSQ-I) and avoidance behaviors manifested by their children after treatment (HSQ-A).

## 4. Discussion

In this study, we demonstrated that a well-known behavioral approach, that is, BST, could be efficiently offered to children with ASD and their parents using a low-cost and commercial web-based training approach. We found a significant reduction in the frequency of behavioral disorders detected in children with ASD after treatment, as well as a general reduction in psychological distress and the perception of the severity of ASD-related symptoms in parents. 

This study demonstrated the feasibility of live coaching on BST procedures as a valid approach to managing behavioral disorders in children with ASD and for helping their parents deal with such disorders during COVID-19 pandemic. Distance behavioral training programs should be encouraged, and public awareness should be raised among parents and clinicians aimed at regarding telehealth as an alternative and valid means of providing treatment [22,23]. In the future, such programs could help tackle the lack of therapists available to support the growing demand for their services and the role of primary caregivers as a critical component in the success of treatments [24,25].

In our study, all the children showed a marked progressive reduction in the frequency of their targeted behavioral symptoms during the follow-up period. In fact, after 20 days (T1), the lowest reduction was 17% and the median was 35%; after 40 days, the lowest was 38% and the median 65%; and finally, at the end of the protocol (60 days), the lowest was 75% (ASD S2) with a median reduction of 90%, with two out of eight children no longer showing the target inappropriate behavior. 

We also found a reduction in parent-reported stress, a decrease in child inflexibility and avoidance, and a more functional parent–child interaction. The only scale that showed no significant reduction (and showed a slight increase) was PSI-DC. This scale is designed to assess the parent’s perception of child-related disorders. To explain this finding we propose that, although our remote BST parent training might induce a better understanding of the difficulties they are facing, the adaptation to stressful events associated with caring for an autistic depends on several factors (family demands; family adaptive resources, the family’s definition of the stressful situation and family adaptive coping mechanisms) [23]. 

There are several studies demonstrating the effectiveness of parental training on the behavioral patterns of children with ASD [26,27]. In fact, parent training can effectively help in improving parent–child interactions and social communication [28]. A study by Tonge et al. [29] demonstrated that both parental education and training in behavioral procedures for children with autism can be beneficial for parental mental health. In Tonge’s study [29], parents received a manual-based education and behavior management skills training package. Sessions were skills-based and action-oriented through the provision of workbooks, modeling, videos, rehearsal, homework tasks, and feedback. At a 6-month follow-up evaluation, the combination of parent education and behavioral teaching led to a significant reduction in anxiety, insomnia, and somatic symptoms. Hassan et al. [30] demonstrated the effectiveness of BST in developing parental skills and treatment strategies to support specific social behaviors in their children. In Hassan’s study, free-play sessions with parents and their children were structured. These authors demonstrated that treatment carried out according to the BST procedure improved functional skills and communication by decreasing the manifestation of destructive behaviors. 

A recent systematic review provided evidence of the utility of telehealth as a service delivery model for providing analytic-based services and for training caregivers to implement behavioral assessments and procedures [31]. Despite this evidence, there are a few studies which evaluated the feasibility of telehealth for families with ASD. Boutain et al. [32] evaluated the success of a remote web-based BST training package to teach parents to implement new treatments with their children for the independent completion of three self-care skills (washing hands, washing faces, and applying lotions). Another study on online coaching in daily living skills, with four children ranging in age from 5 to 9 years, showed that parents faithfully implemented treatment which led to increases in independent daily living skills for all the participants [33]. In line with all this evidence, we demonstrated that a remote BST service is useful in managing behavioral disorders and improving parent awareness by reducing the frequency of various behavioral disorders in children, as well as lowering psychological distress in parents. 

### Limitations

The first limitation of this study was the high attrition rate (42%). Nevertheless, it should be noted that all the drop-outs happened at the beginning of the study and none dropped after the initial assessment. Even if we found no significant difference between drop-outs and families who continued till the end of the protocol, it could be noted that the families who dropped out tended to have working mothers with more children and lower educational levels. This agrees with the concerns expressed during the interviews about the difficulties in reconciling BST with family and work schedules. This is further reinforced by the fact that in all families that continued with the protocol, even if both parents participated in the training, only the mothers actively reported behavioral data. As a consequence, we cannot account for the effect of the involvement and support of the fathers in the therapeutic outcome. Furthermore, we did not record other undesired behaviors, and we therefore do not know whether parents independently applied the learned procedures to other behaviours and if that could partly explain the decrease observed in parental psychological measures. Future studies should improve the feasibility of this procedure and study its contextual needs and accommodations to remotely intercept different populations and family organizations.

Another limitation of the current study was the limited number of participants. A larger sample could provide a more accurate analysis of the effects of treatment. One internal limitation was the lack of a control group which prevented a direct comparison with another type of intervention. Furthermore, the analysis was performed on a group of heterogeneous behavioral disorders and procedures. This limitation is evident due to the large between-subject effect size of the study. Nevertheless, this limitation is also a strength, given the lack of online behavioral studies on the topic. In fact, we believe we have shown the feasibility of the training for a wide set of procedures and hope that this will foster a larger variety of new studies that no longer focus exclusively on a single procedure. 

Finally, we gathered only the frequency of undesired behaviours to avoid overwhelming the parents with too many tasks. Therefore, we did not know of the occurrence of alternative behaviours or the emergence of new undesired behaviours. We recommend that future studies include a larger number of participants and a parallel control group receiving a traditional behavioral treatment. In addition, future research could include an analysis of treatments with homogeneous behaviors. Another important topic would be to assist practicing clinicians in determining an appropriate protocol for the delivery of a 1:1 telehealth service. Some proposals are already in place, but still lack experimental support [31].

## 5. Conclusions

The COVID-19 pandemic has created an enormous amount of suffering for all levels of society, but especially for more vulnerable people. At the same time, it has also led researchers to search for new solutions and to challenge long-held assumptions regarding therapy. Remote health services can provide a sustainable model for both conducting assessments and training healthcare professionals and staff in order to implement several behavioral strategies, as proposed by Ferguson et al. [34]. In this study, we confirm that parental coaching can also be carried out through an easily accessible remote service aimed at making the parents of children with ASD increasingly competent in the daily management of their children and being more empowering in their role. Such remote services can provide support through skills and behavior management treatments not only during global health crises, but also as a sustainable and standard model for the future.

## Figures and Tables

**Figure 1 jcm-11-01194-f001:**
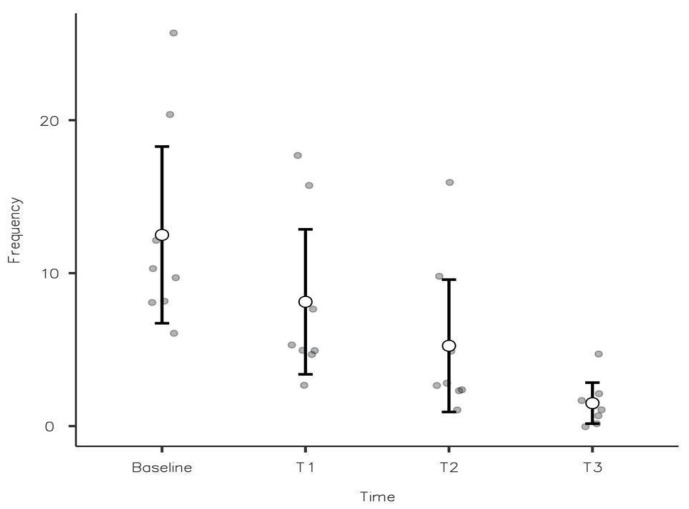
Significant reduction in the frequency of behavioral symptoms during web-based treatment with the BST approach.

**Table 1 jcm-11-01194-t001:** Protocol structure.

Phases	Therapist’s Tasks	Parent’s Tasks
Phase 1	Protocol explanation, Data collection	Behaviours frequency checklist
Phase 2	First baseline, selection of target behaviour, instruction for video recording	Selection of target behaviour, starts video recording
Phase 3	Insert examples in the ABC worksheet from gathered observations. Gives Instruction on ABC worksheet and functional analysis	ABC worksheet recording
Phase 4	Analysis of the type of problem behavior, instruction on functional analysis trough ABC worksheet and videos	Receives instruction, starts the protocol and gather the objective baseline (T0)
Phase 5	Gather T0. Start BST parent training program, teaches procedures for behavioural change	Receive BST
Phase 6	Debrief and fidelity	Receive feedback
Phase 7	Data analysis; external review of fidelity	-
Phase 8	Debrief	Gives feedback

**Table 2 jcm-11-01194-t002:** Timetable of the protocol and data acquisition.

Protocol (days)	Treatment (days)	Week	Phase	Meetings	Data Gathering
-	-	-	0	0	Psychological Assessment T0
0	-	1	1	1	
7	-	2	2	2	
14	-	3	3	3	
21	-	4	4	4	
22	0	-	-	-	Behavioral T0
28	6	5	5	5	
35	13	6	5–6	6	
42	20	7	5–6	7	Behavioral T1
49	27	8	5–6	8	
56	34	9	5–6	9	
62	40	-	-	-	Behavioral T2
63	41	10	5–6	10	
70	48	11	5–6	11	
77	55	12	5–6	12	
82	60	-	-	-	Behavioral T3
83–90	-	13	7	-	
91	-	13	8	13	Psychological Assessment T1

**Table 3 jcm-11-01194-t003:** Demographic and psychological characteristics of children with ASD and their parents.

Measure	Completed	Dropped
Number of children/parents	8/16	6/12
Gender (M/F)	6/2	6/0
Age (months)	72.0 ± 30.4 68 (40–138)	57 ± 14.1 51 (48–84)
Total DQ Griffiths	67.0 ± 19.3 67.5 (45.5–95.5)	63.9 ± 14.2 69.4 (39.0–75.0)
Age of Mother (years)	41.0 ± 5.8 39 (33–52)	37.7 ± 2.3 38 (35–40)
Age of Father (years)	48.4 ± 5.0 49 (41–57)	43.0 ± 4.8 49 (41–57)
Education of Mother (years)	16.1 ± 2.6 18 (13–18)	13.8 ± 2.0 13 (13–18)
Education of Father (years)	16.8 ± 2.3 18 (8–21)	14.7 ± 2.6 18 (8–21)
Working Mother/Father (ratio)	0.50/1.00	0.83/1.00
Number of siblings	0.25 ± 0.46 0 (0–1)	1.33 ± 1.37 1 (0–3)

Data are given as mean values (SD), and median (range). ASD: autism spectrum disorders; DQ: Developmental Quotient. Data are expressed as mean ± SD or median (range) values if assumptions of normality are proved or otherwise.

**Table 4 jcm-11-01194-t004:** Problem behaviors and relative function in the ASD children enrolled.

	Problem Behavior	Behavioral Function	Context
ASD S1	Difficulties in accepting Stop Signal (“no”)	Access to the tangible	At home when routine activities in the presence of both parents are changed or interrupted
ASD S2	Repetitive requests	Access to the tangible	At home during routine activities in the presence of both parents
ASD S3	Unshared laughter	Seeking attention	At home when the child carries out independent activities requested by the parents
ASD S4	Shouting when faced with a task proposed by the mother	Task avoidance/escape	At home when following the mother’s request to perform tasks
ASD S5	Climbing on furniture, shouting, taking dangerous objects	Attention-seeking	At home in the presence of both parents engaged in other activities
ASD S6	Echolalia	Automatic reinforcement	At home when engaged in independent play activities
ASD S7	Throwing objects	Automatic reinforcement	At home while carrying out independent activities, with the mother engaged in other activities
ASD S8	Expressing denial and rejection as a for of idiosyncratic behavior	Task avoidance	At home when following requests by both parents

**Table 5 jcm-11-01194-t005:** Problem procedures and relative explanation in the ASD children enrolled [21].

	Behavioral Procedure	Explanation
ASD S1	Extinction	Parents physically remove themselves from the child when the target behaviour occurs (consequence intervention).
ASD S2	Desensitization	Gradual reduction of the number of requests the child can make and to which the parents can respond (antecedent intervention) according to the established criterion.
ASD S3	Differential reinforcement/Extinction	Extinction: Parents ignore the child when he/she emits the target behavior (consequence intervention) DRO (Differential Reinforcement of Other Behavior): parents were instructed to provide a reinforcement, agreed during the session, whenever the child did not exhibit the problem behavior in a given period of time.
ASD S4	High-p/fading of the prompt	Rapid presentation of a high-probability prompt followed by a low-probability prompt (antecedent intervention). Fading of the prompt: gradual reduction of the help provided by the parents on the task with low probability of the issue (intervention on the antecedent)
ASD S5	Differential reinforcement/Extinction	Extinction: Parents ignore the child when they display the target behavior (consequence intervention) DRO: parents were instructed to provide reinforcement, agreed during the session, whenever the child did not exhibit the problem behavior in a given period of time.
ASD S6	Expanding interests/Direction of Attention	Associating disliked objects/activities with liked, albeit restricted, objects/activities (antecedent intervention)
ASD S7	Expanding interests	Presenting functional auditory stimuli through songs or videos
ASD S8	Token economy	Child had the opportunity to earn tokens during the day (tangible tokens). After reaching an agreed number of tokens, they had the opportunity to exchange them for a reward they liked. The token economy was built on a billboard on which the child could attach points, thus favoring the visual channel for collecting points, and did not use a response cost.

**Table 6 jcm-11-01194-t006:** Frequency of behavioral symptoms during active treatment.

	Main Behavioral Symptoms	Frequency * T0	Frequency T1 (20 days)	Frequency T2 (40 days)	Frequency T3 (60 days)
ASD S1	Difficulties in accepting Stop Signal (“no”)	6	5	2	0
ASD S2	Repetitive requests	20	16	10	5
ASD S3	unshared’ laughter	10	5	5	1
ASD S4	Shouting when faced with a task proposed by the mother	10	5	2	1
ASD S5	Climbing on furniture, shouting, and taking dangerous objects	12	8	3	2
ASD S6	Echolalia	26	18	16	2
ASD S7	Throwing objects	8	3	1	1
ASD S8	Expressing denial and rejection as a for of idiosyncratic behavior	8	5	3	0

* Frequency is expressed as the number of events during the session.

**Table 7 jcm-11-01194-t007:** Psychological effects on parents before and after BST treatment.

	Before Treatment	After Treatment	W/*p*-Level	Paired *t*-Test/*p*-Level
	**PSI-SF Scale**			
Total Value	105.7 ± 7.3 105 (90–114)	95.4 ± 8.1 98 (84–107)	105/0.001	6.2/<0.001
PD	35.6 ± 7.9 36 (21–47)	29.3 ± 8.1 30 (16–43)	67/0.03	2.7/0.02
P-CDI	36.2 ± 8.7 33 (26–52)	29 ± 5.4 29.5 (19–38)	73.5/0.008	3.6/0.003
DC	32.9 ± 12.4 30 (19–53)	36.9 ± 7.1 37 (21–50)	n.s	n.s
	**HSQ-ASD Scale**			
Total Value	4.8 ± 1.8 4.6 (1.9–7.4)	3.5 ± 1.6 3 (1–7)	66/0.003	6.3/<0.001
Inflexibility	5.3 ± 1.6 5.4 (2–8.2)	3.4 ± 1.2 3.3 (2–5)	78/0.002	4.3/0.001
Avoidance	4.8 ± 2.1 4.4 (1.8–7.8)	3.2 ± 2 2.8 (1–6)	91/0.001	7.9/<0.001

Data are given as mean values (SD), and median (range). PSI-SF: Parenting Stress Index/Short Form; HSQ-ASD: Home Situation Questionnaire; PD: Parenting Distress; P-CDI: Dysfunctional parent–child interaction; DC: Difficult Child.

## Data Availability

The datasets generated during the current study are available from the corresponding author on reasonable request.

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
