# Peer review of "Psychological Interventions for Children with Autism during the COVID-19 Pandemic through a Remote Behavioral Skills Training Program"

_jcm, 2022, doi:10.3390/jcm11051194_

Round 1
Reviewer 1 Report
Thank you for the opportunity to review the manuscript, “Skills improvement in children with autism during COVID pandemic through a remote behavioral skills training program.” I have been coaching parents remotely via telehealth technology to implement behavioral interventions for their children with autism and other neurodevelopmental disorders for over five years, thus was pleased to see others embarking on this work to meet the needs of families, particularly during the COVID pandemic. Although I approached this review with enthusiasm, a number of factors preclude me from being able to recommend publication of the manuscript in its current form. Below I outline the primary factors in the order in which they appear in the manuscript.
First, inconsistencies among the title, the stated aim of the investigation, and the findings, the authors’ purpose is unclear. Specifically, the paper’s title suggested the purpose was to use BST to improve autistic children’s skills, the authors stated that “our aim was to validate a new web-based training approach aimed at improving parents’ …competence related to children-related behavioral dysfunctions,” and in the first two sentences of the Discussion section, the authors stated, “…we demonstrated that … BST could be efficiently offered…using a low-cost and commercial web-based training approach…We found a significant reduction in the frequency of behavioral disorders detected.” Thus, the primary dependent variable is unclear (child skill improvement, child behavior reduction, parent competency, or efficient delivery of BST) and the independent variable is unclear (BST is not a new web-based approach, nor is it commercially available). Clarity and consistency regarding the purpose, primary dependent and independent variables is required in any future version of the manuscript.
The authors did not provide sufficient details in the method section to permit replication of their procedures or confidence that the intervention was responsible for the observed data. The following are examples of information that was missing or inadequate.
- The authors stated that they provided parents with instruction in function-based treatments and functional analysis. However, there is no evidence that the treatments selected were function-based (no operant functions were specified). Further, no data were provided on parent behavior, thus it is unknown the degree to which the parents implemented the treatment procedures with fidelity.
- There was no experimental control demonstrated in the within-subjects design, therefore it is unknown whether the observed behavior change was due to parent-implemented intervention or one of several threats to internal validity.
- The authors stated “The objective of the study was to reduce the frequency of problem behaviors by increasing the exhibition of alternative behaviors.” Yet the authors do not provide any data on the occurrence of alternative behavior.
- The authors provided virtually no information regarding the background, experience, or level of training/expertise the behavior therapists possessed.
- The description of what the therapists and parents did during “phases” of the parent training procedures was very confusing. Several of the phases did not appear to involve skill training for the parents, but rather behavior observation, data collection, and intervention selection. If therapists trained parents to conduct each of the steps of a functional assessment process, that should be explicitly stated. Again, the authors did not provide parent behavior data, so there is no way to tell what the parents did during parent training procedures.
- Procedures for collecting and coding child behavior data during intervention are lacking in specifics required for replication.
- The authors did not describe procedures for determining/computing interobserver agreement nor did they report interobserver agreement data, thus the reliability of the observational data are unknown. Unfortunately, this particular flaw renders the data virtually useless.
The authors reported that six parents did not complete the entire web-based protocol. The reader is left to wonder why so many parents did not complete the study, the implications of such a large proportion of the participants failing to complete the study, and whether their reasons for not completing the study might be related to the procedures.
The authors drew conclusions about their findings, such as “this study confirms the feasibility of live coaching on BST procedures” and “...BST as a valid approach to managing behavior…” that are not supported by their data. Specifically, when more than 25% of the participants do not complete the protocol for unstated reasons, the feasibility of the procedures is suspect. Similarly, the authors did not collect or report data on parent behavior or procedural fidelity, so there is no evidence of the validity of BST.
I regret that my recommendation is not in favor of publication. I applaud the authors for their ambitious effort to address the critical needs of families who do not have access to behavioral expertise or support, either due to geographic barriers or to pandemic-related restrictions. I encourage the authors to consider the importance of others being able to replicate their procedures to both strengthen the empirical evidence supporting the procedures and to provide effective support to families of children with autism. In so doing, the authors should take care to collect interobserver agreement and procedural fidelity data, use an experimental design that demonstrates a functional relation between the independent and dependent variables, and provide a detailed description of all procedures.
Author Response
Thank you for the opportunity to review the manuscript, “Skills improvement in children with autism during COVID pandemic through a remote behavioral skills training program.” I have been coaching parents remotely via telehealth technology to implement behavioral interventions for their children with autism and other neurodevelopmental disorders for over five years, thus was pleased to see others embarking on this work to meet the needs of families, particularly during the COVID pandemic. Although I approached this review with enthusiasm, a number of factors preclude me from being able to recommend publication of the manuscript in its current form. Below I outline the primary factors in the order in which they appear in the manuscript.
- First, inconsistencies among the title, the stated aim of the investigation, and the findings, the authors’ purpose is unclear. Specifically, the paper’s title suggested the purpose was to use BST to improve autistic children’s skills, the authors stated that “our aim was to validate a new web-based training approach aimed at improving parents’ …competence related to children-related behavioral dysfunctions,” and in the first two sentences of the Discussion section, the authors stated, “…we demonstrated that … BST could be efficiently offered…using a low-cost and commercial web-based training approach…We found a significant reduction in the frequency of behavioral disorders detected.” Thus, the primary dependent variable is unclear (child skill improvement, child behavior reduction, parent competency, or efficient delivery of BST) and the independent variable is unclear (BST is not a new web-based approach, nor is it commercially available). Clarity and consistency regarding the purpose, primary dependent and independent variables is required in any future version of the manuscript.
REPLY: Title, abstract, introduction and discussion have been changed following this valuable suggestion.
The authors did not provide sufficient details in the method section to permit replication of their procedures or confidence that the intervention was responsible for the observed data. The following are examples of information that was missing or inadequate.
- The authors stated that they provided parents with instruction in function-based treatments and functional analysis. However, there is no evidence that the treatments selected were function-based (no operant functions were specified). Further, no data were provided on parent behavior, thus it is unknown the degree to which the parents implemented the treatment procedures with fidelity.
REPLY: WE would like to thank this reviewer for this important suggestion. We now explain more explicitly in the paper that the functional analysis was carried mainly by the therapists on data gathered by the parents. The procedure we implemented to assess procedural fidelity was explained more carefully and results are now reported in the result section. We rewrote the different phases and added two new tables with the timeline, the actions taken by the different actors and so on. We added the function of each behaviour and for each procedure.
We will not report specific lines because following the reviewer comments we did a major rewriting of the Materials and Methods section.
2. There was no experimental control demonstrated in the within-subjects design, therefore it is unknown whether the observed behavior change was due to parent-implemented intervention or one of several threats to internal validity.
REPLY: We understand the reviewer's perplexity and we agree that it is a limitation of the study (see also the next question). Nevertheless, we did not expect problem behaviours to decrease so strongly and coherently among different timesteps and different subjects due to chance. Obviously having a control group would be better, but repeated measures ANOVA is suited to be used in cases when you have different subjects and different time-steps also without a control group. It’s unlikely that threats to the internal validity will move in the same direction among different subjects and different time steps.
3. The authors stated “The objective of the study was to reduce the frequency of problem behaviors by increasing the exhibition of alternative behaviors.” Yet the authors do not provide any data on the occurrence of alternative behavior.
REPLY: Thanks for noticing, we didn’t use alternative behaviors for each child. Study aim and design are now stated more clearly: “The objective of the study was to assess the effectiveness of tele-assisted BST for parents in reducing the frequency of specific problem behaviors in children with ASD. To this end, given the wide behavioral variability and the absence of a control group a between-group design was not feasible, and a multiple single-subject design was also not feasible due to the wide range of techniques used, difficulty in assessing protocol adherence, fidelity and gathering data daily. Therefore, we applied a repeated measures design with four-time steps, using the frequency of undesired behavior as the outcome variable.”
4. The authors provided virtually no information regarding the background, experience, or level of training/expertise the behavior therapists possessed.
REPLY: Titles and experience are now reported in the paper (at the end of procedures section): “The therapists who delivered the interventions were all chartered psychologists, or psychotherapists, with behavioral analyst training and at least 5 years experience in working with children on the autism spectrum.”
5. The description of what the therapists and parents did during “phases” of the parent training procedures was very confusing. Several of the phases did not appear to involve skill training for the parents, but rather behavior observation, data collection, and intervention selection. If therapists trained parents to conduct each of the steps of a functional assessment process, that should be explicitly stated. Again, the authors did not provide parent behavior data, so there is no way to tell what the parents did during parent training procedures.
REPLY: Thanks for pointing this out, we deeply reformulated this section.
6. Procedures for collecting and coding child behavior data during intervention are lacking in specifics required for replication.
REPLY: We hope that now with the rewriting of different phases the procedure is sufficiently explicit
7. The authors did not describe procedures for determining/computing interobserver agreement nor did they report interobserver agreement data, thus the reliability of the observational data are unknown. Unfortunately, this particular flaw renders the data virtually useless.
REPLY: Now it is better explained in the methods and in the results sections.
8. The authors reported that six parents did not complete the entire web-based protocol. The reader is left to wonder why so many parents did not complete the study, the implications of such a large proportion of the participants failing to complete the study, and whether their reasons for not completing the study might be related to the procedures.
REPLY We agree with the reviewer. The study now reports the reasons and the characteristic of the drop-offs and a comparison table: “Six parents (42%) dropped out before the beginning of phase 1. Those families dropped out of the research due to difficulties in managing weekly online connections, work commitments and/or family management of other children. All the families that started the active treatment phase, completed it until the end of treatment, eight children with ASD and their relative parents (16) were therefore analyzed. Table 3 shows the demographic and psychological characteristics of children with ASD and their parents. Between groups comparison showed no significant differences on any descriptive and psychological characteristic (p>.05).”
9. The authors drew conclusions about their findings, such as “this study confirms the feasibility of live coaching on BST procedures” and “...BST as a valid approach to managing behavior…” that are not supported by their data. Specifically, when more than 25% of the participants do not complete the protocol for unstated reasons, the feasibility of the procedures is suspect. Similarly, the authors did not collect or report data on parent behavior or procedural fidelity, so there is no evidence of the validity of BST.
REPLY: We reworded the conclusion to reduce the hype, from “confirms” to “suggest”. Furthermore, we added the drop-outs discussion in the limitation section:
“The first limitation of the study was the high attrition rate (42%). Nevertheless, it should be noted that all the drop-outs happened at the beginning of the study and none dropped after the initial assessment. Even if we found no significant difference between drop-outs and families who continued till the end of the protocol, it could be noted that the families who dropped out tended to have working mothers with more children and lower educational levels. This agrees with the concerns expressed during the interviews about the difficulties in reconciling BST with the family and work schedule. This is further reinforced by the fact that in all families that continued with the protocol, even if both parents participated in the training, only the mothers actively reported behavioral data. As a consequence, we cannot account for the effect of involvement and support of the fathers in the therapeutic outcome. Furthermore, we didn’t record other undesired behaviors, therefore, we don’t know if parents have independently applied the learned procedures to other behaviours and if that could in part explain the decrease observed in parental psychological measures. Future studies should improve the feasibility of this procedure, and study its contextual needs and accommodations, to remotely intercept different populations and family organizations.“
I regret that my recommendation is not in favor of publication. I applaud the authors for their ambitious effort to address the critical needs of families who do not have access to behavioral expertise or support, either due to geographic barriers or to pandemic-related restrictions. I encourage the authors to consider the importance of others being able to replicate their procedures to both strengthen the empirical evidence supporting the procedures and to provide effective support to families of children with autism. In so doing, the authors should take care to collect interobserver agreement and procedural fidelity data, use an experimental design that demonstrates a functional relation between the independent and dependent variables, and provide a detailed description of all procedures.
REPLY: We would like to express our appreciation for the reviewer’s comments. We feel that our manuscript is strongly improved by incorporating its new suggestions. We replied point-by-point to all suggestions and doubts about the robustness of our study. We hope that the new version of our work would be finally considered for publication.
Reviewer 2 Report
This study evaluated the effectiveness of a web-based parent training program to reduce problem behaviors in children diagnosed with autism spectrum disorder (ASD). Sixteen parents identified a problem behavior in their child and its frequency of occurrence. They then received individualized training from a behavioral consultant with experience administering ABA in client homes. Problem behavior frequency was collected at baseline and 20-, 40-, and 60-days afterward. Parents also completed the Parenting Stress Index – Short Form and Home Situation Questionnaire – ASD before and after the training. Results found a significant reduction in child problem behaviors and parent stress over the study period. The authors conclude that this web-based parent training is an effective way to address behavior problems in children with ASD that can be considered in telehealth models.
This is an important and timely topic given the inevitable changes in healthcare delivery since the onset of the COVID-19 pandemic. Evaluation of telehealth models for children with ASD and other developmental delays and disorders are needed to confirm and improve web-based models of care. However, evaluation of these models must also consider the ability to replicate their effectiveness in different samples of children. In its current form, the paper does not sufficiently describe the training program to facilitate replication. Was a training manual prepared and delivered to the ABA therapists? Did they collect information on problem behaviors in a standardized fashion? Were the same data collection forms used by all participants (e.g., behavior measurement sheet and ABC worksheet)? Can these data collection instruments be made available as an online supplement?
The authors note that the small sample size is a limitation of the study. They should also note the attrition rate as a limitation. Also, were there differences in parents who did and did not complete the training? How could attrition bias influence results?
As stated, 22 parents were enrolled in the study. Six did not complete the training and were excluded. Yet data was presented for only 8 children. The authors should be clear on how many parents completed the training and whether data was collected from one or both parents of each child. If data was collected from both parents, how were discrepancies in reporting resolved?
Further, it is not clear how problems behaviors were chosen as the focal point of parent training. More information should be provided on the process to help parents choose which problem behavior of their child to address.
A few other notes:
- The authors note several times that there was a decrease in behavioral disorders during the study period. Instead, there was a decrease in the frequency of problem behaviors. The authors should revise to make this point clear.
- The authors switch between using the terms autism, ASD and autistic. Please use one term for consistency. I prefer autism spectrum disorder (ASD) to align with the DSM-5.
- The introduction would be strengthened if literature was reviewed on behavior problems that emerged among children with ASD during the COVID-19 pandemic.
- Please clarify whether T1, T2, and T3 measures were taken 20-, 40-, and 60-days after baseline or after the training was completed.
Author Response
This study evaluated the effectiveness of a web-based parent training program to reduce problem behaviors in children diagnosed with autism spectrum disorder (ASD). Sixteen parents identified a problem behavior in their child and its frequency of occurrence. They then received individualized training from a behavioral consultant with experience administering ABA in client homes. Problem behavior frequency was collected at baseline and 20-, 40-, and 60-days afterward. Parents also completed the Parenting Stress Index – Short Form and Home Situation Questionnaire – ASD before and after the training. Results found a significant reduction in child problem behaviors and parent stress over the study period. The authors conclude that this web-based parent training is an effective way to address behavior problems in children with ASD that can be considered in telehealth models. This is an important and timely topic given the inevitable changes in healthcare delivery since the onset of the COVID-19 pandemic. Evaluation of telehealth models for children with ASD and other developmental delays and disorders are needed to confirm and improve web-based models of care. However, evaluation of these models must also consider the ability to replicate their effectiveness in different samples of children. In its current form, the paper does not sufficiently describe the training program to facilitate replication.
REPLY: We would like to express our appreciation for the reviewer’s comments. We feel that our manuscript is strongly improved by incorporating its suggestions. We replied point-by-point to all the reviewers’ suggestions and doubts about the robustness of our study. We hope that the new version of our work would be finally considered for publication.
1. Was a training manual prepared and delivered to the ABA therapists? Did they collect information on problem behaviors in a standardized fashion? Were the same data collection forms used by all participants (e.g., behavior measurement sheet and ABC worksheet)? Can these data collection instruments be made available as an online supplement?
REPLY: There was no training manual delivered to therapists, while data collections forms are now uploaded as supplementary materials.
2. The authors note that the small sample size is a limitation of the study. They should also note the attrition rate as a limitation. Also, were there differences in parents who did and did not complete the training? How could attrition bias influence results?
REPLY: Attrition rate is now reported more clearly in the first part of the results section as well as in the limitations section. Moreover, differences between parents completing and not completing the training have been reported. See Table 3
3. As stated, 22 parents were enrolled in the study. Six did not complete the training and were excluded. Yet data was presented for only 8 children. The authors should be clear on how many parents completed the training and whether data was collected from one or both parents of each child. If data was collected from both parents, how were discrepancies in reporting resolved?
REPLY: Thanks for noticing the typo, now we correctly report that initially there were 28 parents (not 22). The first part of the Results section has been reformulated. Behavioral data were collected only from one parent (it was the mother in all cases who enrolled actively in the therapy). Parent-reports were collected from both parents.
4. Further, it is not clear how problems behaviors were chosen as the focal point of parent training. More information should be provided on the process to help parents choose which problem behavior of their child to address.
REPLY: WE agree with this reviewer that this section should be improved. Indeed the explanation of different phases is now explained in more detail. Specifically Phase 1 to 2:
"In Phase 1, the first operative meeting was scheduled. The therapist collects information on the individual child’s behaviors and discusses them with the parents. At the end of the meeting, the therapist provided them with an individualized frequency checklist in order to have an operationalized definition of the problem behaviors reported by the parents. Identified behaviour should be observable, measurable, and repeatable. In the following week the parents observed and monitored their child’s problem behaviors whilst completing the behavior frequency checklist. In Phase 2, the second operative meeting was scheduled. The therapist and the parents discussed the results reported in the checklist and selected the target behaviour. During the online meeting, the therapist established a first baseline related to the problem behavior. The behavior selected for treatment was the more frequently observed behavior that reduced learning opportunities or social inclusion or was either physically or emotionally harmful for the child or their family. At the end of the meeting, the parents were instructed to record videos of the target behavior during the following week.”
A few other notes:
- The authors note several times that there was a decrease in behavioral disorders during the study period. Instead, there was a decrease in the frequency of problem behaviors. The authors should revise to make this point clear.
Reply: Modified
- The authors switch between using the terms autism, ASD and autistic. Please use one term for consistency. I prefer autism spectrum disorder (ASD) to align with the DSM-5.
Reply: Modified
- The introduction would be strengthened if literature was reviewed on behavior problems that emerged among children with ASD during the COVID-19 pandemic.
Reply: Done
- Please clarify whether T1, T2, and T3 measures were taken 20-, 40-, and 60-days after baseline or after the training was completed.
Reply: T1-T2 and T-3 started after training. Specifically, phase 5. This additional information has been included in the methods section.
Round 2
Reviewer 1 Report
The revised paper is greatly improved. I do not believe that others would be able to replicate these procedures given the lack of clarity of procedural detail and some references to ABA are a bit misleading. Nonetheless, practitioners in clinical medicine could benefit from this if they were to consult with a behavior analyst for implementation.
This manuscript is a resubmission of an earlier submission. The following is a list of the peer review reports and author responses from that submission.
Round 1
Reviewer 1 Report
Thank you for the opportunity to review the manuscript titled “Promoting skill improvement in children with autism during the COVID pandemic through a remote health service based on a behavioral skill training program”. This is certainly a timely manuscript and describes the transition to telehealth that many behavioral therapists and psychologists have taken since the pandemic began. There are a number of major and minor issues that I had with the manuscript that have led me to recommend rejection. Ultimately, my biggest concerns are that this paper does not add anything new to the research literature and if it does this was not adequately communicated in the paper. There is a great deal of detail that has been left out that made it difficult to evaluate the merit of this study. For example, the use of raw frequency data without any description of the contexts the data were collected in or the time element is simply too hard to overlook. My comments and suggestions are outlined below and are broken down by manuscript section:
Introduction:
No major concerns.
Minor Concerns:
Page 1, Lines 42-43: The end of this sentences does not make much sense, particularly “…promoting new organizational measures.” Please clarify.
Page 2, Lines 52-53: What is meant by “psychological distance”?
Page 2, Line 56: This cites CASP, but the reference in the Reference section is for the Council on Children with Disabilities.
Page 2, Line 66: The reference cited here (10: Lerman et al., 2008) is neither related to parents, nor telehealth/remote health service.
Materials and Methods:
Major Concerns:
Page 3 – Functional Analysis. The description included here is a descriptive analysis and should be labeled as such. Most often, the term “functional analysis” is used to describe experimental analysis procedures where environmental events are manipulated to identify the events likely to evoke problem behavior and the variables that maintain/reinforce problem behavior (e.g., Iwata et al., 1994).
Page 3 – Choice of Target Behaviors. I don’t think that this section is necessary, but I do think it is necessary to indicate how the target behaviors were chosen. For example, did parents nominate the target behaviors or did a therapist observe and nominate behaviors?
Page 4 – BST Procedures. This section describes the components of BST, but does not make explicit what procedures were actually implemented. For example, Lines 160-161 indicate that BST instructions may be “…oral or written, in both cases they should be clear, short and limited in number”. In this study, were they oral, written, both? This section should be very clear about the procedures conducted and not focused on generally defining BST.
Page 4 – Outcome Measures. There is a major issue with the use of targeted problem behavior as an outcome measure without any description of the contexts where the data were collected. It may be that some contexts did not allow for problem behavior to occur or the establishing operation associated with the problem behavior was not in place (e.g., a child who displays aggression when told “no” to accessing a preferred item would be unlikely to display that behavior if he/she was not restricted during data collection). This does not allow for adequate assessment of the child’s problem behavior or changes resulting from the treatment.
Page 4 - The HSQ should not be considered a measure of parent psychological well-being. It is a measure of child behavior and cannot be conflated with parent well-being.
Minor Concerns:
Page 3 – Inclusion criteria. Please explain what “ability of receiving in-home rehabilitation service” means.
I suggest eliminating the adjective “inadequate” and using “inappropriate” or “problem” throughout.
I suggest different terms than “anamnestic”, “telematic”, “cards” (use “worksheet”),
Page 3, Line 137. Descriptive assessment does not need to be capitalized.
Results:
Major Concerns:
Page 6 – Table 2. It is unclear what most of the data in this table mean. There is no indication what the data represent (e.g., mo., yr. IQ scores, et.). The note under the table does not appear to match the table at all.
Once again, the raw frequency data regarding child problem behavior is not helpful without understanding the contexts or whether the opportunity (e.g., the EO was present) existed.
Minor Concerns:
Page 5 – It looks like both parents for each child were included in the study. I did not see any confirmation that both parents participated in all sessions. Were there cases where both parents were not available?
Page 6 – Table 3. Although all of these behaviors may be problematic and interfering, given many of the recent criticisms to ABA therapy related to treating “autistic behaviors”, it would be wise to indicate why behaviors such as echolalia, continuous requesting, and “non-contextualized laughter” would be targets for intervention.
I assume that “desensibilization” under the behavior procedures for continues requests is actually “desensitization”. If not, I am not familiar with that procedure and have never seen it in the literature.
Discussion
Major Concerns:
Page 8. Line 287. I do not believe that telehealth is an independent variable in research anymore. It is the modality of delivery and the true IV should be the treatment of interest (e.g., BST in this study).
Page 9. Line 313. This is the first reference to a manual or workbooks. Nothing regarding a manual or workbooks was described in the method section. This should have been included.
Line 363-365. “The aim of this research was to create a useful, easily accessible, and low-cost service to make parents of children with ASD increasingly competent in the daily management of their children and empowered in their role.” There are many aspects of this aim that were not evaluated. For example, there is no measure on social validity related to the ease of use or access, or the cost associated with it.
Minor Concerns:
Line 291. I’m not sure what “lower” kind of treatment is referring to. Is this referring to BST or the use of telehealth?
Line 299. I would avoid calling the behavior extinguished as this may not be the mechanism resulting in the reduction of the behavior.
Author Response
Thank you for the opportunity to review the manuscript titled “Promoting skill improvement in children with autism during the COVID pandemic through a remote health service based on a behavioral skill training program”. This is certainly a timely manuscript and describes the transition to telehealth that many behavioral therapists and psychologists have taken since the pandemic began. There are a number of major and minor issues that I had with the manuscript that have led me to recommend rejection. Ultimately, my biggest concerns are that this paper does not add anything new to the research literature and if it does this was not adequately communicated in the paper. There is a great deal of detail that has been left out that made it difficult to evaluate the merit of this study. For example, the use of raw frequency data without any description of the contexts the data were collected in or the time element is simply too hard to overlook. My comments and suggestions are outlined below and are broken down by manuscript section:
Introduction:
No major concerns.
Minor Concerns:
- Page 1, Lines 42-43: The end of this sentences does not make much sense, particularly “…promoting new organizational measures.” Please clarify.
REPLY: This sentence has been re-formulated
- Page 2, Lines 52-53: What is meant by “psychological distance”?
REPLY: the term “psychological” has been eliminated
- Page 2, Line 56: This cites CASP, but the reference in the Reference section is for the Council on Children with Disabilities.
REPLY: This reference has been modified accordingly
- Page 2, Line 66: The reference cited here (10: Lerman et al., 2008) is neither related to parents, nor telehealth/remote health service.
REPLY: A new reference has been added
Materials and Methods:
Major Concerns:
- Page 3 – Functional Analysis. The description included here is a descriptive analysis and should be labeled as such. Most often, the term “functional analysis” is used to describe experimental analysis procedures where environmental events are manipulated to identify the events likely to evoke problem behavior and the variables that maintain/reinforce problem behavior (e.g., Iwata et al., 1994).
REPLY: the Description of Functional analysis has been modified accordingly
- Page 3 – Choice of Target Behaviors. I don’t think that this section is necessary, but I do think it is necessary to indicate how the target behaviors were chosen. For example, did parents nominate the target behaviors or did a therapist observe and nominate behaviors? Page 4 – BST Procedures. This section describes the components of BST, but does not make explicit what procedures were actually implemented. For example, Lines 160-161 indicate that BST instructions may be “…oral or written, in both cases they should be clear, short and limited in number”. In this study, were they oral, written, both? This section should be very clear about the procedures conducted and not focused on generally defining BST.
REPLY: We would like to thank this reviewer for these very important suggestions. The entire parenting training procedure (including choice of target behaviors & BST procedures) has been re-written following the reviewer’s suggestions.
- Page 4 – Outcome Measures. There is a major issue with the use of targeted problem behavior as an outcome measure without any description of the contexts where the data were collected. It may be that some contexts did not allow for problem behavior to occur or the establishing operation associated with the problem behavior was not in place (e.g., a child who displays aggression when told “no” to accessing a preferred item would be unlikely to display that behavior if he/she was not restricted during data collection). This does not allow for adequate assessment of the child’s problem behavior or changes resulting from the treatment.
REPLY: The entire 2.6 paragraph describing outcome measurement has been re-written following the reviewer’s suggestions.
- Page 4 - The HSQ should not be considered a measure of parent psychological well-being. It is a measure of child behavior and cannot be conflated with parent well-being.
REPLY: Following the reviewer’s suggestion, we better re-formulated this section: “As concerns parents, we collected outcome measures on parental stress together with a scale useful for assessing the parent's perception of the child's behavioral manifestations. This evaluation was made in two different timepoints at baseline and at the end of treatment. The main outcome measures used were the Parental Stress Index (PSI) to assess the level of stress before and after treatment and the Home Situation Questionnaire (HSQ-ASD), which provides objective measures of perception and influence of the behavior of children in the life of parents”
Minor Concerns:
- Page 3 – Inclusion criteria. Please explain what “ability of receiving in-home rehabilitation service” means.
REPLY: Indeed, this inclusion criteria has been removed
- I suggest eliminating the adjective “inadequate” and using “inappropriate” or “problem” throughout.
REPLY: Done
- I suggest different terms than “anamnestic”, “telematic”, “cards” (use “worksheet”),
REPLY: Done
- Page 3, Line 137. Descriptive assessment does not need to be capitalized.
REPLY: DONE
Results:
Major Concerns:
- Page 6 – Table 2. It is unclear what most of the data in this table mean. There is no indication what the data represent (e.g., mo., yr. IQ scores, et.). The note under the table does not appear to match the table at all.
REPLY: Table 2 has been modified
- Once again, the raw frequency data regarding child problem behavior is not helpful without understanding the contexts or whether the opportunity (e.g., the EO was present) existed.
REPLY: Table 3 has been modified accordingly including the description of the context.
Minor Concerns:
- Page 5 – It looks like both parents for each child were included in the study. I did not see any confirmation that both parents participated in all sessions. Were there cases where both parents were not available?
REPLY: Now described in 2.5 section
- Page 6 – Table 3. Although all of these behaviors may be problematic and interfering, given many of the recent criticisms to ABA therapy related to treating “autistic behaviors”, it would be wise to indicate why behaviors such as echolalia, continuous requesting, and “non-contextualized laughter” would be targets for intervention.
REPLY: As better specified in the new version of the 2.5 section, we considered targets for intervention, problem behavior proposed by the parents as those frequently observed reducing learning opportunities or social inclusion or are either physically or emotionally harmful for the child or their family
- I assume that “desensibilization” under the behavior procedures for continues requests is actually “desensitization”. If not, I am not familiar with that procedure and have never seen it in the literature.
REPLY: Indeed, changed
Discussion
Major Concerns:
- Page 8. Line 287. I do not believe that telehealth is an independent variable in research anymore. It is the modality of delivery and the true IV should be the treatment of interest (e.g., BST in this study).
REPLY: Following the reviewer’s suggestion, we formulated this conclusion.
- Page 9. Line 313. This is the first reference to a manual or workbooks. Nothing regarding a manual or workbooks was described in the method section. This should have been included.
REPLY: The sentence beginning with “In this study, parents received a manual-based…” refers to the Tonge et al., work. We modified this statement to avoid misleading interpretations.
- Line 363-365. “The aim of this research was to create a useful, easily accessible, and low-cost service to make parents of children with ASD increasingly competent in the daily management of their children and empowered in their role.” There are many aspects of this aim that were not evaluated. For example, there is no measure on social validity related to the ease of use or access, or the cost associated with it.
REPLY: This statement has been re-formulated accordingly.
Minor Concerns:
- Line 291. I’m not sure what “lower” kind of treatment is referring to. Is this referring to BST or the use of telehealth?
REPLY: Modified
- Line 299. I would avoid calling the behavior extinguished as this may not be the mechanism resulting in the reduction of the behavior.
REPLY: This statement has been changed in: “with 2 out of 8 children achieving a significative reduction in behavior emission frequency.”
Reviewer 2 Report
Thank you for the opportunity to review the manuscript: “Promoting skill improvement in children with autism during the COVID pandemic through a remote health service based on a behavioral skill training program.” I believe this is a very important and timely topic given the impact of COVID on child mental health and the delivery of behavioral services to children with autism spectrum disorder (ASD) and their families.
I appreciated the authors’ focus in the introduction about the benefits of telehealth, including its utility for things such as service access, continuity of services, and clinical opportunities to foster relationships and conduct naturalistic observations as it relates to working with children and families. From what I can tell from reading, the specific focus of the study was using telehealth to deliver behavioral skill training (BST) to caregivers and measuring effectiveness via two components: 1. change in child target behaviors, and 2. change in parent-reported self-ratings of emotional and psychological constructs, including sense of competency and stress from pre-treatment to post-treatment. I believe these outcome domains make sense and are consistent with the outcome measures of other studies investigating caregiver-mediated behavior interventions.
The authors thoroughly described their inclusion criteria for parent participants. I did question a few of them, regarding age (any rationale for the upper age limit for parents?) and the need for “availability of receiving in-home rehabilitation service” (given that the study intervention was conducted remotely). For the child inclusion criteria, the authors note that the children had a diagnosis of ASD which was further confirmed through evaluation and consent by a research team member. How was this confirmed (standardized measure, clinical interview, rating scales)?
For the study design, the authors state their primary objective (reduction of “inadequate behaviors”) and their within-subjects design, as it is noted that no control group was included (and is noted as a limitation later on). I wonder about the use of the term, “inadequate behaviors.” It seems this is being used to designate challenging behaviors which are the targets of intervention. To me, inadequate seems to imply “not enough” of something, similar to a skill deficit, rather than the presence of undesired behaviors. I wonder if a clarification or different term may help the reader understand that these “inadequate” behaviors are the challenging/undesired behaviors that the intervention aims to decrease. Perhaps giving more examples and operationally defining it more in the term’s first use would help (first said around line 107). This is done later (around line 119) which is helpful, but this is not the first time this term is used.
Reading through the procedure, the authors describe 8 “weekly meetings” and then move into talking about “phases.” I think each “phase” constitutes 1 “weekly meeting” but I think this could be said more clearly to get a better sense of the intervention itself. Additionally, I am not sure I am clear on what each session looked like. Who were the clinicians? Who attended the sessions (caregiver, child, others)? Eventually, the authors talk about the interventions used with each child/parent (e.g., extinction). Who selected the interventions? Was it prescriptive or did families choose/have say in the behavior procedures? Also, the procedures were only mentioned in Table 3. Were these the procedures that parents role-played with clinicians (per the text)? What was feedback like – Coaching? Video review? Visual inspection of data? I think additional details could help the reader better understand the intervention approach and what was involved, since it is the main component of this study.
Related to the behaviors and the analyses, the authors make claims such as “behavior’s frequency should be high enough to measure change during the session” (line 124-125) and “on the basis of our experience in treating behavioral disorders in young children with ASD, we expect very large effects. For high frequency target behavior we expect a reduction of 50%-100%...; lines 213-215). Where do these claims come from and what support is there for expected effects? Later on in the Table 4, the baseline frequencies of behaviors range from 6 to 26 – are these all considered “high enough” to measure change given the different nature of the behaviors targeted? Is the “high frequency” threshold for something like echolalia the same as something like “launching items”?
Moving into the results, I have questions regarding some of the table content. For Table 2, I am unclear how the footnotes relate to the table. There are references to measures like STAI-Y, BDI-II, but these aren’t referenced anywhere else. Additionally, what are the two rows of numbers in Age (months) and Total QS Griffith? Each of these rows includes two rows of numbers which I cannot distinguish from one another. If one is mean (SD) and one is median (range), as is the case for a later table (Table 5), please denote. Also, this is the only place Total QS Griffith is mentioned – what is this? This needs to be defined/explained if it is meant to be there. Similarly, in the table footnote – there is a list of education labels, but I don’t see how this relates to the data included. These labels seem to be more categorical, but education seems to be listed in the sum of years (unclear). In order to understand the demographic data, additional edits and clarification is needed.
Moving into the discussion, I note that the introduction of it talks about “individuals with and without intellectual disabilities” but this is the first time intellectual disability (ID) is mentioned and included. If this is a distinction or note to be made, I think it should be referenced in the paper. IQ was not explicitly talked about, other than in the demographics table (see notes above about questions regarding IQ measure/table). The authors might also consider the overall flow of the paper. The discussion includes studies regarding BST and telehealth literature. I wonder if this would have been more helpful in the literature review to “set the stage” of the paper/study more. The introduction currently talks about COVID and telehealth, but talks little about behavior interventions, which is the focus of the paper.
Overall, I would also strongly recommend the authors review the manuscript with additional editors. Specific attention should be given to grammar and syntax. There are frequent times where terms are not used or spelled consistently (e.g., behavior vs. behaviour), terms are used without being defined/explained (e.g., “ABA” line 129) and/or terms are used that may impact readability (“anamnestic”) or should be considered within the context of disability advocacy and rights (e.g., describing the effects on those in society that are “most fragile” [line 357] or discussing intervention as being done “on” children [lines 73-74]). I believe that there are better ways to describe these constructs/ideas while also being mindful of the parties we are talking about and things like agency.
I believe this study addresses an important topic that is likely to be relevant currently and in the future as telehealth continues to be used during the COVID-19 pandemic and beyond. In order to be publishable, I believe this article requires additional work to address both content and readability.
Author Response
Thank you for the opportunity to review the manuscript: “Promoting skill improvement in children with autism during the COVID pandemic through a remote health service based on a behavioral skill training program.” I believe this is a very important and timely topic given the impact of COVID on child mental health and the delivery of behavioral services to children with autism spectrum disorder (ASD) and their families. I appreciated the authors’ focus in the introduction about the benefits of telehealth, including its utility for things such as service access, continuity of services, and clinical opportunities to foster relationships and conduct naturalistic observations as it relates to working with children and families. From what I can tell from reading, the specific focus of the study was using telehealth to deliver behavioral skill training (BST) to caregivers and measuring effectiveness via two components: 1. change in child target behaviors, and 2. change in parent-reported self-ratings of emotional and psychological constructs, including sense of competency and stress from pre-treatment to post-treatment. I believe these outcome domains make sense and are consistent with the outcome measures of other studies investigating caregiver-mediated behavior interventions.
REPLY: We would like to thank this reviewer for these kind comments.
- The authors thoroughly described their inclusion criteria for parent participants. I did question a few of them, regarding age (any rationale for the upper age limit for parents?)
REPLY: We would like to thank this reviewer for highlighting this typo. We removed it. Parental age was not an inclusion criteria
- and the need for “availability of receiving in-home rehabilitation service” (given that the study intervention was conducted remotely).
REPLY: This inclusion criterion has been reformulated.
- For the child inclusion criteria, the authors note that the children had a diagnosis of ASD which was further confirmed through evaluation and consent by a research team member. How was this confirmed (standardized measure, clinical interview, rating scales)?
REPLY: Clinical diagnosis has been improved and re-formulated
- For the study design, the authors state their primary objective (reduction of “inadequate behaviors”) and their within-subjects design, as it is noted that no control group was included (and is noted as a limitation later on). I wonder about the use of the term, “inadequate behaviors.” It seems this is being used to designate challenging behaviors which are the targets of intervention. To me, inadequate seems to imply “not enough” of something, similar to a skill deficit, rather than the presence of undesired behaviors. I wonder if a clarification or different term may help the reader understand that these “inadequate” behaviors are the challenging/undesired behaviors that the intervention aims to decrease. Perhaps giving more examples and operationally defining it more in the term’s first use would help (first said around line 107). This is done later (around line 119) which is helpful, but this is not the first time this term is used.
REPLY: We agree with this reviewer. The term “inadequate” has been removed throughout the paper.
- Reading through the procedure, the authors describe 8 “weekly meetings” and then move into talking about “phases.” I think each “phase” constitutes 1 “weekly meeting” but I think this could be said more clearly to get a better sense of the intervention itself. Additionally, I am not sure I am clear on what each session looked like. Who were the clinicians? Who attended the sessions (caregiver, child, others)? Eventually, the authors talk about the interventions used with each child/parent (e.g., extinction). Who selected the interventions? Was it prescriptive or did families choose/have say in the behavior procedures? Also, the procedures were only mentioned in Table 3. Were these the procedures that parents role-played with clinicians (per the text)? What was feedback like – Coaching? Video review? Visual inspection of data? I think additional details could help the reader better understand the intervention approach and what was involved, since it is the main component of this study.
REPLY: We would like to thank this reviewer for this very important suggestion. The entire parental training procedure (including Interventions and Outcomes measurements sections) has been re-written also following suggestions required by reviewer n°1.
- Related to the behaviors and the analyses, the authors make claims such as “behavior’s frequency should be high enough to measure change during the session” (line 124-125) and “on the basis of our experience in treating behavioral disorders in young children with ASD, we expect very large effects. For high frequency target behavior we expect a reduction of 50%-100%...; lines 213-215). Where do these claims come from and what support is there for expected effects? Later on in the Table 4, the baseline frequencies of behaviors range from 6 to 26 – are these all considered “high enough” to measure change given the different nature of the behaviors targeted? Is the “high frequency” threshold for something like echolalia the same as something like “launching items”?
REPLY: This section has been completely reformulated in agreement with the reviewer’s suggestions
- Moving into the results, I have questions regarding some of the table content. For Table 2, I am unclear how the footnotes relate to the table. There are references to measures like STAI-Y, BDI-II, but these aren’t referenced anywhere else. Additionally, what are the two rows of numbers in Age (months) and Total QS Griffith? Each of these rows includes two rows of numbers which I cannot distinguish from one another. If one is mean (SD) and one is median (range), as is the case for a later table (Table 5), please denote. Also, this is the only place Total QS Griffith is mentioned – what is this? This needs to be defined/explained if it is meant to be there. Similarly, in the table footnote – there is a list of education labels, but I don’t see how this relates to the data included. These labels seem to be more categorical, but education seems to be listed in the sum of years (unclear). In order to understand the demographic data, additional edits and clarification is needed.
REPLY: We would like to thank this reviewer for highlighting these typos in Table 2. A more coherent version has been included. As concerns data presentation, we confirm that following Kolmogorov–Smirnov analysis, only psychological measures recorded on parents are characterized by a gaussian distribution. Following the reviewer’s suggestion, we modified section 2.7 and table 2.
- Moving into the discussion, I note that the introduction of it talks about “individuals with and without intellectual disabilities” but this is the first time intellectual disability (ID) is mentioned and included. If this is a distinction or note to be made, I think it should be referenced in the paper. IQ was not explicitly talked about, other than in the demographics table (see notes above about questions regarding IQ measure/table).
REPLY: The first statement in the discussion has been removed
- The authors might also consider the overall flow of the paper. The discussion includes studies regarding BST and telehealth literature. I wonder if this would have been more helpful in the literature review to “set the stage” of the paper/study more. The introduction currently talks about COVID and telehealth, but talks little about behavior interventions, which is the focus of the paper.
REPLY: The overall flow of this paper is thought for highlighting in the first part of the introduction the impact of the COVID-19 pandemic on the daily management of children with ASD. Next, we moved to explain the opportunities offered by Telemedicine to provide continuous treatment at home. We believe that the meaning and the role of behavioral interventions on ASD children and parents should be more stressed in the discussion when we compare our results with respect to other remote parent training treatments.
- Overall, I would also strongly recommend the authors review the manuscript with additional editors. Specific attention should be given to grammar and syntax. There are frequent times where terms are not used or spelled consistently (e.g., behavior vs. behaviour), terms are used without being defined/explained (e.g., “ABA” line 129) and/or terms are used that may impact readability (“anamnestic”) or should be considered within the context of disability advocacy and rights (e.g., describing the effects on those in society that are “most fragile” [line 357] or discussing intervention as being done “on” children [lines 73-74]). I believe that there are better ways to describe these constructs/ideas while also being mindful of the parties we are talking about and things like agency.
“and/or terms are used that may impact readability (“anamnestic”) or should be considered within the context of disability advocacy and rights (e.g., describing the effects on those in society that are “most fragile” [line 357] or discussing intervention as being done “on” children [lines 73-74]). I believe that there are better ways to describe these constructs/ideas while also being mindful of the parties we are talking about and things like agency”
RE: As suggested by this reviewer we performed an overall correction of terms, typos, and statements throughout the paper.
I believe this study addresses an important topic that is likely to be relevant currently and in the future as telehealth continues to be used during the COVID-19 pandemic and beyond. In order to be publishable, I believe this article requires additional work to address both content and readability.